Ecological interactions and the Netflix problem

Desjardins-Proulx Philippe philippe.d.proulx@gmail.com 1 2
Laigle Idaline 1
Poisot Timothée 2
Gravel Dominique 1
1 Université de Sherbrooke , Sherbrooke , Quebec , Canada
2 Université de Montréal , Montréal , Quebec , Canada
Orlov Yuriy
Electronic publication date: 2017 Aug 10
Publication date: 2017
Volume: 5
Electronic Location ID: e3644
Received 2017 Apr 25; Accepted 2017 Jul 12
Copyright: ©2017 Desjardins-Proulx et al.
Copyright year: 2017
Copyright holder: Desjardins-Proulx et al.
License: This is an open access article distributed under the terms of the Creative Commons Attribution License, which permits unrestricted use, distribution, reproduction and adaptation in any medium and for any purpose provided that it is properly attributed. For attribution, the original author(s), title, publication source (PeerJ) and either DOI or URL of the article must be cited.
License URL: https://creativecommons.org/licenses/by/4.0/

Keywords: Food web, Ecology, Species interactions

Funding: National Sciences and Engineering Research Council of Canada Microsoft NVIDIA Canada Research Chair program NSERC Discovery grant FQRNT Nouveau Chercheur grant PDP was funded by an Alexander Graham Bell Graduate Scholarship from the National Sciences and Engineering Research Council of Canada, an Azure for Research award from Microsoft, and benefited from the Hardware Donation Program from NVIDIA. DG is funded by the Canada Research Chair program and NSERC Discovery grant. TP is funded by an NSERC Discovery grant and an FQRNT Nouveau Chercheur grants. The funders had no role in study design, data collection and analysis, decision to publish, or preparation of the manuscript.

==============================
Species interactions are a key component of ecosystems but we generally have an incomplete picture of who-eats-who in a given community. Different techniques have been devised to predict species interactions using theoretical models or abundances. Here, we explore the K nearest neighbour approach, with a special emphasis on recommendation, along with a supervised machine learning technique. Recommenders are algorithms developed for companies like Netflix to predict whether a customer will like a product given the preferences of similar customers. These machine learning techniques are well-suited to study binary ecological interactions since they focus on positive-only data. By removing a prey from a predator, we find that recommenders can guess the missing prey around 50% of the times on the first try, with up to 881 possibilities. Traits do not improve significantly the results for the K nearest neighbour, although a simple test with a supervised learning approach (random forests) show we can predict interactions with high accuracy using only three traits per species. This result shows that binary interactions can be predicted without regard to the ecological community given only three variables: body mass and two variables for the species’ phylogeny. These techniques are complementary, as recommenders can predict interactions in the absence of traits, using only information about other species’ interactions, while supervised learning algorithms such as random forests base their predictions on traits only but do not exploit other species’ interactions. Further work should focus on developing custom similarity measures specialized for ecology to improve the KNN algorithms and using richer data to capture indirect relationships between species.

Introduction

Species form complex networks of interactions and understanding these interactions is a major goal of ecology (Pimm, 1982). The problem of predicting whether two species will interact has been approached from various perspectives (Bartomeus et al., 2016; Morales-Castilla et al., 2015). Williams & Martinez (2000) for instance built a simple theoretical model capable of generating binary food webs sharing important features with real food webs (Gravel et al., 2013), while others have worked to predict interactions from species abundance data (Aderhold et al., 2012; Canard et al., 2014) or exploiting food web topology (Cohen, 1978; Staniczenko et al., 2010). Being able to predict with high enough accuracy whether two species will interact given simply two sets of attributes, or the preferences of similar species, would be of value to conservation and invasion biology, allowing us to build food webs with partial information about interactions and help us understand cascading effects caused by perturbations. However, the problem is made difficult by the small number of interactions relative to non-interactions and relationships that involve more than two species (Golubski et al., 2016).

In 2006, Netflix offered a prize to anyone who would improve their recommender system by more than 10%. It took three years before a team could claim the prize, and the efforts greatly helped advancing machine learning methods for recommenders (Murphy, 2012). Recommender systems try to predict the rating a user would give to an item, recommending them items they would like based on what similar users like (Aggarwal, 2016). Ecological interactions can also be described this way: we want to know how much a species would “like” a prey. Interactions are treated as binary variables, two species interact or they do not, but the same methods could be applied to interaction matrices with preferences. There are two different ways to see the problem of species interactions. In the positive-only case, a species has a set of preys, and we want to predict what other preys they might be interested in. This approach has the benefit of relying only on our most reliable information: positive (preferably observed) interactions. The other approach is to see binary interactions as a matrix filled with interactions (1s) and non-interactions (0s). Here, we want to predict the value of a specific missing entry (is species xi consuming species xj?). For this paper, we focus on the positive-only approach, which relies on a simple machine learning approach called the K nearest neighbour.

Statistical machine learning algorithms (Murphy, 2012) have proven to be reliable to build effective predictive models for complex data (the “unreasonable effectiveness of data” Halevy, Norvig & Pereira, 2009). The K nearest neighbour (KNN) algorithm is an effective and simple algorithm for recommendation, in this case finding good preys to a species with positive-only information. The technique is simple: for a given species, we find the K most similar species according to some distance measure, and use these K species to base a prediction. If all the K most similar species prey on species x, there is a good chance that our species has interest in x. In our case, similarity is simply computed using traits and known interactions, but more advanced techniques could be used with a larger set of networks. For example, it is possible to learn similarity measures instead of using a fixed scheme (Bellet, Habrard & Sebban, 2015). For this study, we use a data-set from Digel et al. (2014), which contains 909 species, of which 881 are involved in predator—prey relationships and 871 have at least one prey. The data comes from soil food webs and includes invertebrates, plants, bacteria, and fungi. In total, the data-set has 34,193 interactions. The data was complemented with information on 25 binary attributes (traits) for each species, plus their body mass and information on their phylogeny. We also compare our approach to a supervised learning method, random forests, which is used to predict interactions with only the species’ traits.

A summary of the two methods used can be found in Table 1. The approaches are not directly comparable. For example, the positive-only KNN recommends preys to a species. If we remove a prey from a species, ask the algorithm to recommend a prey, and check whether the prey will come up as the recommendation, there are up to 881 possibilities. On the other hand, the random forest predicts either an interaction or a non-interaction, a 50% chance of success by random. These approaches have different uses. Positive-only algorithms are interesting because we are rarely certain that two species do not interact. Also, the KNN approach uses information on what similar species do, while random forests only rely on traits.

Table 1 Summary of the two methods used.

The recommender uses the K nearest neighbour algorithm with the Tanimoto distance measure. The Tanimoto KNN makes a recommendation, while supervised learning with random forests (RF) predict either an interaction or a non-interaction.

Method	Input	Prediction	
Recommender (KNN)	Set of traits & preys for each species	Recommend new preys	
Supervised learning (RF)	Traits (binary and real-valued)	Interaction (1) or non-interaction (0)	

We show the KNN is particularly effective at retrieving missing interactions in the positive-only case, succeeding 50% of the times at recommending the right species among 881 possibilities. With few traits, the random forests can achieve high accuracy (≈98% for both interactions and non-interactions) without any information about other species in the community. Random forests require only three traits to be effective: body mass and two traits based on the species’ phylogeny. Our results show that, with either three traits per species or partial knowledge of the interactions, it is possible to reconstruct a food web accurately. These approaches are complementary, covering both the case where traits are readily available and when only partial knowledge of the food web is known. Both techniques can be used to reconstruct a food web with different types of information.

Method

Data

The first data-set was obtained from the study of Digel et al. (2014), who documented the presence and absence of interactions among 881 species from 48 forest soil food webs, details of which are provided in the original publication. A total of 34,193 unique interactions were observed across the 48 food webs, and a total of 215,418 absence of interactions. In order to improve representation of interactions involving low trophic levels species that were not identified at the species level in the first data-set, we compiled a second data-set from a review of the literature. We selected all articles involving interactions of terrestrial invertebrate species for a total of 126 studies, across these, a total of 1,439 interactions were recorded between 648 species. Only 88 absences of interactions were found. We selected traits based on to their potential role in consumption interactions (Table 2). For each species or taxa, these traits were documented based on a literature review or from visual assessment of pictures. In addition to these traits, we included two proxies for hard-to-measure traits: feeding guild and taxonomy. The traits were chosen for their potential relevance for species interactions and their availability, see (Laigle et al., in press) for greater details on the data-set.

Table 2 The traits used.

All traits are binary except for body mass, Ph0, and Ph1. We use taxonomy as a proxy of latent traits following Mouquet et al. (2012). To do so, we used the R package ape to obtain taxonomic distances between the species, perform classical multidimensional scaling (or principal coordinates analysis) (Cox & Cox, 2001) on taxonomic distances, and use the scores of each species on the first two axes (Ph0 and Ph1) as taxonomy-based traits. These three real-valued variables are scaled to be in the [0, 1) range. For the Tanimoto similarity index, these three continuous variables have to be converted to binary features. For each, we create four binary features of equal size (n = 881∕4).

Features	Abbr.	Description	n	
AboveGroud	AG	Whether the species live above the ground.	538	
Annelida	An	For species of the annelida phylum.	34	
Arthropoda	Ar	For species of the arthropoda phylum.	813	
Bacteria	Bc	For species of the bacteria domain.	1	
BelowGround	BG	For species living below the ground.	464	
Carnivore	Ca	For species eating other animals.	481	
Crawls	Cr	Whether the species crawls.	184	
Cyanobacteria	Cy	Member of the cyanobacteria phylum.	1	
Detritivore	De	For species eating detribus.	355	
Detritus	Ds	Whether the species can be classifying as a detritus.	2	
Fungivore	Fg	For species eating fungi.	111	
Fungi	Fu	Member of the fungi kingdom.	2	
HasShell	HS	Whether the species has a shell.	274	
Herbivore	He	For species eating plants.	130	
Immobile	Im	For immobile species.	85	
IsHard	IH	Whether the species has a though exterior (but not a shell).	418	
Jumps	Ju	Whether the species can jump.	30	
LongLegs	LL	For species with long legs.	59	
Mollusca	Mo	Member of the mollusca phylum.	45	
Nematoda	Ne	Member of the nematoda phylum.	5	
Plantae	Pl	Member of the plant kinggom.	3	
Protozoa	Pr	Member of the protozoa kingdom.	3	
ShortLegs	SL	For species with short legs.	538	
UsePoison	UP	Whether the species uses poison.	177	
WebBuilder	WB	Whether the species builds webs.	89	
Body mass	M	Natural logarithm of the body mass in grams	881	
Ph0	Ph0	Coordinate on the first axis of a PCA of phylogenetic distances	881	
Ph1	Ph1	Coordinate on the second axis of a PCA of phylogenetic distances	881	

K-nearest neighbour

Our recommender uses the K-nearest neighbour (KNN) algorithm (Murphy, 2012). The KNN algorithm is an instance-based method, it does not build a general internal model of the data but instead bases predictions on the K nearest (i.e., most similar) entries given some distance metrics. In the case of recommendation, each species is described by a set of traits and a set of preys, and the algorithm will recommend new preys to the species based on the preys of its K nearest neighbours. For example, if K = 3, we take the set of preys of the three most similar species to decide which prey to recommend. If species A is found twice and B once in the set of preys of the most similar species, we will recommend A first (assuming, of course, that the species does not already have this prey). See Table 3 for a complete example of recommendation. In the “Netflix” problem, this is equivalent to recommending new TV series/movies to a user by searching for the users with the most similar taste and using what they liked as recommendation. It is also possible to tackle the reverse problem: Amazon uses item-based recommendations, in which case we are looking for similar items instead of similar users to base our recommendations (Aggarwal, 2016).

Table 3 Fictional example to illustrate recommendations with K nearest neighbour using the Tanimoto distance measure modified to include species traits.

We are trying to recommend a prey to species 0 given that the three most similar species are species 6, 28, and 70. For example, the distance from species 0 to species 70 would be wt0.5 + (1 − wt)2∕4. To find recommendations, the set of preys found in the K = 3 most similar entries is computed, in this case {812 = 2, 70 = 2, 72 = 1}, leading to the list of recommendations [812, 70, 72]. Because they are found most often in the K most similar species, candidates 812 and 70 will be suggested before 72. To test this approach, we remove a prey from a species and check whether the algorithm recommend the missing prey. Especially with low K, it’s possible that no recommendations can be found, for example if the most similar species has the exact same preys.

Species ID	Traits	Preys	Most similar	Recommendations	
0	{Ar, Ca}	{6, 42, 47}	{6, 28, 70}	[812, 70, 72]	
6	{Ar, Ca}	{42, 47, 70, 72}			
28	{Ar, Ca}	{42, 47, 70, 812}			
70	{Ca}	{42, 47, 812}			
…	…	…			

Choosing the right value for K is tricky. Low values give high importance to the most similar entries, while high values provide a larger set of examples. Fortunately, the most computationally intensive task is to compute the distances between all pairs, a step that is independent of K. As a consequence, once the distances are computed, we can quickly run the algorithm with different values of K.

Different distance measures can be used. We will use the Tanimoto coefficient for recommendations. The Tanimoto (or Jaccard) similarity measure is defined as the size of the intersection of two sets divided by their union, or: (1) tanimotox,y=x∩yx∪y,

Since it is a similarity measure in the [0, 1] range, we can transform it into a distance function with 1 − tanimoto(x, y). The distance function uses two types of information: the set of traits of the species (see Table 2) and their set of preys. We define the distance function with traits as: (2) tanimotodx,y,wt=wt1−tanimotoxt,yt+1−wt1−tanimotoxi,yi,

where wt is the weight given to traits, xt and yt are the sets of traits for species x and y, and xi, yi are their sets of preys. Thus, when wt = 0, only interactions are used to compute the distance, and when wt = 1, only traits are used. See Table 3 for an example.

The data is the set of preys and binary traits for each species (Table 2). To test the approach, we randomly remove an interaction for each species and ask the algorithm to recommend up to 10 preys for the species with the missing interaction. Interactions are removed one-at-a-time and similarity is computed before the interaction is removed. The code for computing similarities after the interaction is removed is available in the code repository, but it has little effect on the results while making the program much slower to run since the similarity matrix must be computed for each trial. We count how many recommendations are required to retrieve the missing interactions and compute the top1, top5, and top10 success rates, which are defined as the probabilities to retrieve the missing interaction with 1, 5, or 10 recommendations. We repeat this process 10 times for each species with at least 2 preys, totally 7,200 attempts. We test all odd values of K from 1 to 19, and wt = {0, 0.2, 0.4, 0.6, 0.8, 1}. We also divided species in groups according to the number of preys they have to see if it is easier to find the missing interaction for species with fewer preys.

Supervised learning

We also do a simple test with random forests to see if it is possible to predict interactions in this data-set using only the traits (Breiman, 2001). In this case, the random forests perform supervised learning: we are trying to predict y (interaction) from the vector of traits x by first learning a model on the training set, and testing the learned model on a testing set. We keep 5% of the data for testing. We perform grid search to find the optimal parameters for the random forests.

For our predictions, we count the number of true positives (tp), true negatives (tn), false positives (fp) and false negatives (fn). The score for predicting interactions (Scorey), non-interactions (Score¬y) and the accuracy are defined as

(3) Scorey=tptp+fp,

(4) Score¬y=tntn+fn,

(5) Accuracy=Scorey34193+Score¬y7419688812,

with 34,193 and 74,1968 being the number of observed interactions and non-interactions in the 881 by 881 matrix. We then use the True Skill Statistics (TSS) to measure how accurate the random forest is, defined a (6) TSS=tp×tn−fp×fntp+fnfp+tn.

The TSS ranges from −1 to 1.

Code and data

Since several machine learning algorithms depends on computing distances (or similarities) for all pairs, many data structures have been designed to compute them efficiently from kd-trees discovered more than thirty years ago (Friedman, Bentley & Finkel, 1977) to ball trees, metric skip lists, navigating nets (Izbicki & Shelton, 2015), and cover trees (Beygelzimer, Kakade & Langford, 2006; Izbicki & Shelton, 2015). We use an exact but naive approach that works well with small data-sets. Since distance(x, y) = 0 if x = y and distance(x, y) = distance(y, x), our C++ implementation stores the distances in a lower triangular matrix without the diagonal, yielding n(n − 1)∕2 distances to compute. A linear scan is then used to find the most similar species. Computing the distance matrix and testing the predictions 7,000 times for a set of parameters takes less than a second. We used Scikit for random forests (Pedregosa et al., 2011). The C++11 code for the KNN algorithm, Python scripts for random forests, and all data-sets used are available at https://github.com/PhDP/EcoInter (also stored on Zenodo with a DOI: Desjardins-Proulx, 2016).

Results

Recommendation

While matrix imputation has a 50% change of success by random, the Tanimoto KNN needs to pick the right prey among up to 881 possibilities. Yet, it succeeds on its first recommendation around 50% of the times. When the first recommendation fails, the next nine recommendations only retrieve the right species around 15% of the times so the top5 and top10 success rates are fairly close to the top1 success rate (see Fig. 1). The Tanimoto measure is particularly effective for species with fewer preys, achieving more than 80% success rate for species with 10 or fewer preys (Fig. 2).

Figure 1 Finding the missing interaction with KNN/Tanimoto approach. After removing a prey from a predator, we ask the KNN algorithms with Tanimoto measure to make 10 recommendations (from best to worst).

The figure shows how many recommendations are required to retrieve the missing interaction. Most retrieved interactions are found with the first attempt. This data was generated with K = 7 and wt = 0.

Figure 2 Success on first guess with Tanimoto similarity as a function of the number of prey.

The KNN algorithm with Tanimoto similarity is more effective at predicting missing preys when the number of preys is small. This is probably in good part because there are more information available to the algorithm, since 473 species have 10 or fewer preys, 295 have between 10 and 100, 103 species have more than 100 preys.

The highest first-try success rates (the probability to pick the missing interaction on the first recommendation) are found with K = 7 and no weights to traits, and with K = 17 and a small weight of 0.2 to traits (Fig. 3). Overall, the value of K had little effect on predictive ability.

Figure 3 Top1 success rates for the KNN/Tanimoto algorithm with various K and weights to traits.

When wt = 0.0, the algorithm will only use interactions to compute similarity between species. When wt = 1, the algorithm will only consider the species’ traits (see Table 2). The value is the probability to retrieve the correct missing interaction with the first recommendation. For each entry, n = 871 (the number of species minus 10, the number of species with no preys). The best result is achieved with K = 17 and w = 0.2, although the results for most values of K and w = [0.0, 0.2] are all fairly close. The success rate increases with K when only traits are considered (w = 1).

Supervised learning

Random forests predict correctly 99.55% of the non-interactions and 96.81% of the interactions, for a TSS of 0.96. Much of this accuracy is due to the three real-valued traits (body mass, Ph0, Ph1). Without them, too many entries have the same feature vector x, making it impossible for the algorithm to classify them correctly. Removing the binary traits has little effect on the model. With only body mass, Ph0, Ph1, the TSS of the random forests is 0.94.

Discussion

We applied different machine learning techniques to the problem of predicting binary species interactions. Recommendation is arguably a better fit for binary species interactions, since it is essentially the same problem commercial recommenders such as Netflix face: given that a user like item i, what is the best way to select other items the user would like? In this case, users are species, and the items are their preys, but the problem is the same. In both cases, we can have solid positive evidence (observed or implied interactions), but rarely have proofs of non-interactions. The approach yields strong results, with a top1 success rate above 50% in a food web with up to 881 possibilities. The approach could be used, for example, to reconstruct entire food webs using global database of interactions (Poelen, Simons & Mungall, 2014). The method’s effectiveness rely on nestedness: how much species cluster around the same set of preys in a food web (Guimaraes & Guimaraes, 2006). Thus, it should be less effective in food webs with more unique predators.

The KNN algorithm falls into the realm of unsupervised learning, where the goal is to find patterns in data (Murphy, 2012). The other class of machine learning algorithms, supervised learning, have the clearer goal of predicting a value y from a vector of features x. For example, in supervised learning, we would try to predict an interaction y from the vector of traits x, while a unsupervised approach can fill entries from an incomplete matrix regardless of what the entry is (interaction or trait). With a larger set of food webs, it may be possible to use an unsupervised algorithm, for example a neural network, to train a model for matrix imputation. Instead of recommending new preys, imputation would simply fill missing entries from a matrix (interaction or non-interactions).

Our random forests achieve a TSS of 0.96 using the binary traits, body mass, and the coordinates of the multidimensional scaling. This is consistent with previous research that has shown that ecological networks have relatively few dimensions (Eklof et al., 2013). A random forest can build effective predictive models by creating complex rules based on the traits, while the KNN algorithm relies on a simplistic distance metrics. However, the KNN approach has some advantages over supervised learning, namely the capacity to recommend preys using only the information from the other species’ interactions. The solution to improve the KNN approach in ecology is likely to learn distance metrics (Bellet, Habrard & Sebban, 2015) instead of using a fixed formula. This would allow complex rules while maintaining the KNN’s ability to exploit partial food web structures. Learning distance metrics is a promising avenue to improve our results. Much efforts on the Netflix prize focused on improving similarity measures (Toscher & Jahrer, 2008; Hong & Tsamis, 2006), and custom similarity metrics can be used to improve unsupervised classification algorithms (Bellet, Habrard & Sebban, 2015) by exploiting complex domain-specific rules. Maybe species with many preys, apex predators, or specialists behave differently enough to need different similarity measures. Learning distance metrics from data is a common way to improve methods based on a nearest neighbour search (Xing et al., 2003; Bellet, Habrard & Sebban, 2015), allowing the measure itself to be optimized. We only used the K nearest neighbour algorithm for unsupervised learning, but several other algorithms can be used to solve the “Netflix problem”. For example: techniques based on linear programming, such as recent exact methods for matrix completion based on convex optimization (Candès & Recht, 2009) or low-rank matrix factorization. The latter method reduces a matrix to a multiplication between two smaller matrices, which can be used both to predict missing entries and to compress large matrices into small, more manageable matrices (Vanderbei, 2013). Given enough data, deep learning methods such as deep Boltzmann machines could also be used (Zhang, 2011). Deep learning revolutionized machine learning with neural networks made of layers capable of learning increasingly detailed representations of complex data (Hinton, Osindero & Teh, 2006). Many of the most spectacular successes of machine learning use deep learning (Mnih et al., 2013). However, learning several neural layers to form a deep networks would require larger data-sets.

The low sensitivity to K in recommendations is interesting and makes the approach easier to use. This is caused by the fact that, as K grows, the set of species includes more and more unrelated species with widely different set of preys. If we increase K from k to k + δ for a recommendation, the species in δ range are not only less similar, but they are less likely to share preys among themselves. Since recommendations are based on how many times a prey is found in the K nearest species, the species in the δ range are unlikely to have as much weight as the first k species. Our KNN recommender is particularly easy to parametrize since it is neither sensible to the weight given to traits nor to the choice of K.

Our results have two limitations. It is possible that our food web was exceptionally simple, and that a food web with distinct structural properties would behave differently, especially if it has lower nestedness. The success of the KNN algorithms depends on local structure: how much can we learn from similar species. If each species has a unique set of preys, the KNN will struggle more. Also, a deeper issue is that real food webs are not binary structures. Species, populations, and individuals have different densities, prey more strongly on some resources than others, and have preferences. In a binary matrix, we can predict if two species will interact while completely ignoring the rest of the network, but real food webs involve complex indirect relationships (Wootton, 1994). It is unclear how much we can learn about ecosystems and species interactions from binary matrices, and our results show that binary interactions can be predicted without direct knowledge of the community, since we are able to effectively predict if two species interact given only three traits. Species interactions are better represented with a weighted hypergraph (Gao et al., 2012), which is well-suited to model relations with an arbitrary number of participants. The hyperedge would allow for complex indirect relationships to be included. Understanding these hypergraphs is outside the scope of the KNN algorithm but could be understood with modern techniques such as Markov logic (Richardson & Domingos, 2006).

Recommendation (KNN algorithm with Tanimoto distance) and supervised learning (random forests) are complementary techniques. Supervised learning is more useful when we have traits and no information about interactions, but it is useless without the traits. On the other hand, the recommender performs well without traits but requires at least partial information about interactions, although it might be possible to use the interactions from different food webs. We suggest more research could be done on developing better distance metrics for ecological interactions or learning these metrics from data.

We thank Anna Eklof and one anonymous reviewer for helpful comments.

Additional Information and Declarations

Competing Interests

Author Contributions

Data Availability

The authors declare there are no competing interests.

Philippe Desjardins-Proulx conceived and designed the experiments, performed the experiments, analyzed the data, contributed reagents/materials/analysis tools, wrote the paper, prepared figures and/or tables, reviewed drafts of the paper.

Idaline Laigle contributed reagents/materials/analysis tools, reviewed drafts of the paper, formatted and organised the data.

Timothée Poisot and Dominique Gravel conceived and designed the experiments, contributed reagents/materials/analysis tools, reviewed drafts of the paper.

The following information was supplied regarding data availability:

The repository https://github.com/PhDP/EcoInter has all the data & code to check the results (doi: 10.5281/zenodo.161602).

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
