# Peer review of "Ecological interactions and the Netflix problem"

_PeerJ, doi:10.7717/peerj.3644_

## Round 0.1 · original submission · Major Revisions

Dear Philippe,

Appended are the two reviews of your manuscript. Please take them into account and revise the text accordingly. I sent you first review several days ago to save time. The second review demands fewer changes in the text. You are welcome to submit an updated manuscript.

Reviewer 1 ·

Basic reporting

Fine.

Experimental design

Fine.

Validity of the findings

Fine.

Additional comments

Review of "Ecological interactions and the Netflix problem" by Desjardins-Proulx et al.

The authors consider two different modelling approaches for predicting biotic interactions in ecological communities. The first is a "recommendation" approach based on the Netflix problem that uses information about the traits and feeding interactions of a focal species' nearest neighbours to predict possible interactions involving the focal species. The second is a supervised learning approach called random forests that uses only information on species' traits to predict possible interactions across the entire community. The authors find that predictions are not significantly improved by including species' traits in the recommendation approach. For the random forest approach, only three traits (body mass and two variables for species phylogeny) were required to predict interactions with high accuracy. The authors conclude by suggesting that the two approaches are complementary, and that further work should focus on developing techniques that use supervised learning to inform a refined subset of species for consideration in recommendation algorithms.

The manuscript is clear and well-written. The idea of using recommendation algorithms for predicting species interactions is new and represents and interesting application of machine learning techniques in ecology. I have one major comment about justifying the inclusion of the supervised machine learning approach, along with suggestions for improvement in this regard. If this comment is addressed, then I would be supportive of publication in PeerJ.


-- Justifying the inclusion of the supervised machine learning approach

The focus of the manuscript is clearly the recommendation approach, which the authors contrast with a supervised learning approach. However, the authors note that the two approaches "are not directly comparable" (P3L71). Although I think it's valuable to compare the recommendation approach to a different approach, this inability to directly compare the two approaches somewhat undermines the motivation for including the random forest analysis in the manuscript. As a reader, I felt like the two approaches were being described and presented in parallel to one another, and for most of my read-through it was not really clear what the link was between the two approaches. I don't think the paper would lose much by cutting the random forest analysis altogether. However, the inclusion of the supervised learning approach (random forests) becomes clearer towards the end of the manuscript, where the authors suggest that "the solution to the [recommendation] approach in ecology is likely to *learn* distance metrics instead of using a fixed formula" (P13L207).

I think the manuscript would benefit greatly from the authors attempting even a simple demonstration of their idea. For example, they could use a combination of body size, Ph0 and Ph1 (the most informative traits from their random forest analysis) to measure distances, then use this learned distance matrix to make prey-set comparisons as in the recommendation approach (i.e., instead of the Tanimoto distance). This would be an interesting analysis in its own right, and would also serve to justify the inclusion of the random forest approach in the manuscript.


-- Minor comments

Spelling or grammar suggestions are prefaced by "s/g:"

P1L10. s/g: "who-eats-whom"

P1L17. Without knowing that there are 882 species in the food web under consideration, it is not immediately clear what the number "881" represents. Also, this sentence needs to be explained more fully, as is done on P3L72. (Side note: Are there 882 or 881 species under consideration? On P3L65, the authors suggest that "881 [species] are involved in predator-prey relationships" but elsewhere it is noted that there are 882 species, e.g., P4L89.)

P2 Introduction. The authors may already be aware of the following article, but another nice example linking popular technology ideas and ecology is Allesina, S. and Pascual, M. (2009) Googling Food Webs: Can an Eigenvector Measure Species’ Importance for Coextinctions? PLOS Comput Biol, 5, e1000494

P3L59. The authors may be interested in citing some other articles that have used only the topology of food webs to determine other possible trophic interactions. Namely: Cohen, J.E. (1978). Food Webs and Niche Space. Princeton University Press, Princeton, NJ (look for the concept of the "resource graph", which, I believe was introduced by George Sugihara); and Staniczenko, P.P.A. et al. (2010) Structural dynamics and robustness of food webs. Ecol Lett, 13, 891-899 (where a similar concept is referred to as the "predator-overlap graph").

P3L62. s/g: "... simply computed using traits ..."

P3L75. s/g: "... the random forest predicts either ..."

P4L103. s/g: "... instead bases predictions ..."

P5 Table 2 caption. Please explain (most likely in the main text) how the "four binary features" are created, especially as they end up being the most informative.

P6L110. s/g: "... equivalent to recommending new TV series ..."

P7 Table 3. I don't fully understand the example presented in the caption of Table 3. Comparing Species 0 to Species 70, the authors suggest that the distance to Species 70 is w_t 0.5 + (1 - w_t) 1/3. By my application of the Tanimoto similarity (Eqn 1), the similarity between the set of traits is 1/2 (ok); but the similarity between the set of preys is 2/4 = 1/2 (Species 42 and 47 are shared, out of the total set of Species 6, 42, 47 and 812). So where does the 1/3 come from (I contend that it should be 1 - 1/2 = 1/2)? Are the authors using the number of preys of Species 0 as the denominator? If so, this needs to be made clear, especially as it is then no longer consistent with the definition of the Tanimoto similarity.

P7L129. "To test the approach, we randomly remove an interaction for each species and ask the algorithm to recommend up to 10 preys for the species with the missing interaction." The authors should add more details. For example, are interactions removed one-at-a-time or all-in-one-go? Is similarity calculated before or after removal?

P14L227. I think the authors mean "sensitivity" rather than "sensibility" here (and similarly on P14L234).

P14L228. s/g: "This is caused by ..."

P14L241. s/g: "... if two species will interact while ..."

P14L244. "... our results show that binary interactions are mostly independent of the community, since we are able to effectively predict if two species interact given only three traits." I think this is an overly strong assertion by the authors. It might well be the case that the traits used to make predictions about binary interactions have been shaped by community factors, especially when the mentioned traits involve taxonomy and phylogeny.

·

Basic reporting

The manuscript is clearly written and is well organised.

I do miss some references, and of course some lines on this, about other approaches that have used species traits in identifying resource species for consumers in food webs and other ecological networks. Even if this manuscript is more of a methods paper this about using traits is such an important part that it deserves some more attention I believe. One example paper is Eklöf et al, 2013, Ecol Lett "The dimensionality of food webs" who used traits of various identities to predict interactions. Then we of course have the large body of
literature using body size/mass

Experimental design

The research question is clearly started and defined and the methods are well described and easy to follow.

However, I am curious of why exactly these traits (listed in Table 2) were chosen? On line 96 it is stated "selected traits based on their potential role in consumption interactions", but to me this is not a good enough explanation. The traits show a huge span in how detailed they are, and some of them I would not even refer to as "traits". For example the taxonomy entities is clearly a proxy for numerous traits, and the same for the phylogenetic distances. (This also opens up for some discussion on using phylogeny in predicting interactions, e.g, work by L-F Bersier and colleagues.) It becomes clear when looking at the results for the random forest analyses the complexity captured by these "traits" as the accuracy only can be achieved with Ph0 and Ph1.

Validity of the findings

One criticism, that the authors also mention themselves, is that it would be interesting to also test this on a different data set - one that is not including only soil food webs (as these tend to be a bit special). However as the paper is a description of the methodology, and not focus on the specific results, it is not urgent to add more data.

Additional comments

I really enjoyed reading this manuscript and found it interesting and sound.

---

## Round 0.2 · accepted · Accept

All the major comments were taken into account in the revised manuscript. Second reviewer had minor remarks that were fixed without second review round. As editor I have no more comments

Reviewer 1 ·

Basic reporting

Fine.

Experimental design

Fine.

Validity of the findings

Fine.

Additional comments

Review of revised "Ecological interactions and the Netflix problem" by Desjardins-Proulx et al.

I am Reviewer 1 from the first round of reviews. I appreciate that the authors attempted my suggestion for using the results of the random forest approach as inputs to the unsupervised "Netfilx" approach, showing that it doesn't work, and I understand their reasoning for why it should not be expected to work. On re-reading their discussion of how the unsupervised approach could be improved by "learn[ing] distance metrics instead of using a fixed formula" (P13L219), it occurred to me that I my have initially misunderstood what the authors were suggesting. Now, I very much like the ideas presented in the discussion on this point, but suggest that readers my benefit from more ecological context, perhaps by expanding on the examples given on P14L224. Finally, I like the new closing to the introduction section (P4L84), which helps with clarifying the motivation for the work. I am happy with all the other changes in the manuscript and am supportive of publication in PeerJ.